# Single-scan detection of ligand-binding using hyperpolarization and low-field relaxation
Pooja Narwal [1,4], Nils Lorz [2,4], Masoud Minaei [1,4], Alvar D. Gossert [2] ✉ & Benno Meier [1,3] ✉

The nuclear spin-lattice relaxation rate $1/T_1$ depends on the correlation time $\tau_c$ of the molecule bearing the nuclear spin, and can therefore probe changes of $\tau_c$ upon binding of a rapidly moving small ligand to a more slowly moving larger protein. In practice however, the dependence is such that only a small difference in relaxation rate is obtained at high field. Here we present a scheme in which nuclear spins are first hyperpolarized using DNP, and then allowed to relax at low magnetic field in presence of a target protein, which generates a large $T_1$ contrast. The sample is subsequently transferred into a conventional nuclear magnetic resonance probe (NMR), where the effect of the low-field relaxation is read out using high-field liquid-state NMR. Using only 14 µM of a $^{13}$C-labeled reporter ligand, we observe protein binding reliably for protein concentrations as low as 2 µM in a single scan. The scheme is expanded to a label-free ligand via a competitive binding experiment in which the label-free ligand displaces the $^{13}$C-labeled reporter ligand.

The observation of ligand-protein binding is a cornerstone of drug discovery and biomolecular research. Nuclear magnetic resonance (NMR) can directly observe this binding process in solution (thus requiring essentially no assay development), and is able to scan thousands of compounds within days[1]. Following hit identification, NMR can be used to assert ligand and protein integrity and determine affinity[2].

The NMR experiments typically used for observing ligand-protein binding are ligand-based and make use of fast exchange processes that are characteristic for weak to moderately strong binders. In the saturation-transfer-difference experiment (STD), saturation of the protein and protein-bound ligands leads, upon exchange, to a reduction of the signal of the free ligand[3]. In the water-LOGSY experiment, the saturation of aqueous protons causes a Nuclear Overhauser Effect (NOE) that differs in sign for the free and bound ligand, causing the net effect of the saturation to be a sensitive probe of binding[4]. In the $T_{1\rho}$ experiment and the closely related CPMG experiment, one exploits enhanced transverse relaxation upon binding[5]. We note that long-lived spin states are often highly sensitive to binding and can be used to detect binding at extremely low protein concentrations. Examples of ligands that exhibit such states are molecules with aliphatic chains[6,7] and alanine-glycine and citrate[8]. In summary, the established experiments take advantage of the NOE or $T_2$ relaxation; and do not make use of $T_1$ relaxation.

Common to all of these experiments is the need to record sufficiently strong ligand signals, which translates to typical ligand concentrations on the order of 200 µM. In turn, these high ligand concentrations imply high protein concentrations on the order of 20 µM to achieve sufficiently strong relaxation effects on the ligand. Since NMR requires relatively large sample volumes compared to other screening techniques, the required protein amount is likewise significant. Relevant protein targets often need to be expressed in delicate, low yield systems, such as insect and mammalian cells[9], making it desirable to reduce the required protein concentration as far as possible.

The recently introduced PEARLScreen still employs high ligand concentrations of 200 µM. By exploiting long $T_2$ relaxation times, the PEARLScreen can detect binding using protein concentrations as low as 1 µM, provided that an efficient exchange relaxation mechanism is active, and that the measurement is carried out at high magnetic field (600 MHz or for even better contrast 1.2 GHz)[10]. Near thermal equilibrium, similar sensitivities have previously only been achieved for molecules bearing aliphatic groups, where long-lived spin states[11] are highly sensitive to binding[6,7]. Nevertheless, while these experiments enable a reduction in protein concentration, the required ligand concentration remains high.

An alternative approach to sensitive binding detection is to depart from thermal equilibrium, and boost the ligand signal using hyperpolarization techniques such as dissolution-dynamic nuclear polarization[12] or SABRE[13]. The larger ligand signals then enable one to work with

[1]Institute of Biological Interfaces 4, Karlsruhe Institute of Technology, Eggenstein-Leopoldshafen, Germany. [2]Department of Biology, ETH Zürich, Zürich, Switzerland. [3]Institute of Physical Chemistry, Karlsruhe Institute of Technology, Karlsruhe, Germany. [4]These authors contributed equally: Pooja Narwal, Nils Lorz, Masoud Minaei. ✉e-mail: alvar.gossert@biol.ethz.ch; benno.meier@kit.edu

smaller ligand concentrations. Consequently, smaller protein concentrations are required to cause a significant effect on the ligand signals. For fluorinated ligands, the group of Hilty has shown that a strong reduction of the fluorine $T_2$ upon reversible binding to a protein enables concentrations as low as 1 μM for both the protein and the ligand[14]. Likewise, the $^{13}C$ $T_2$ can probe affinity in a site-resolved manner, provided that the concentrations are adjusted for the low natural abundance of $^{13}C$[15]. The group of Jannin has recently combined hyperpolarization with benchtop NMR and was able to observe binding of hyperpolarized ligands at a concentration of 600 μM using a protein concentration of 40 μM[16]. Several molecules, in particular derivatives of tyrosine and tryptophan can be hyperpolarized and used to observe binding in aqueous solution at ambient temperature via the photo-CIDNP effect[17,18].

It is well known that not only $T_2$, but also $T_1$ depends on correlation time[19]. Specifically, $T_1$ depends on the available spectral density at 0, 1 and 2 times the Larmor frequency. Since the spectral density is itself a function of correlation time, so is $T_1$. Random-field relaxation depends only on the spectral density at the Larmor frequency, and is given by:

$$R_1 = T_1^{-1} = \gamma^2 B_r^2 \frac{\tau_c}{1 + (\omega \tau_c)^2} \qquad (1)$$

where $\gamma$ is the gyromagnetic ratio of the nucleus, $B_r$ is the root-mean-square amplitude of the random field and $\tau_c$ is the correlation time. The dependence of the $^{13}C$ relaxation rate on correlation time is shown in Fig. 1A. As can be seen, the difference in relaxation rate between a free ligand (assuming a correlation time of 100 ps) and a ligand bound to a protein (assuming a correlation time of 10 ns) is less than 3 at 9.4 Tesla. However, the same difference exceeds 50 when the magnetic field is reduced to a value of the order of 1 Tesla. This value is close to the theoretical limit of 100 which is obtained in the limit of $\omega^2\tau_C \ll 1$. We note that a similar dependence of relaxation rate on correlation time is obtained for dipole-dipole relaxation. Other relaxation mechanisms (such as chemical shift anisotropy) are assumed to be negligible.

Based on these considerations, we designed a hyperpolarization experiment that maximizes the available $T_1$ contrast. In the experiment (Fig. 1B), the ligand is first hyperpolarized, and then transferred to a liquid-state NMR magnet where it is dissolved in a solution at room temperature. The solution contains the protein at a defined concentration and, critically, is kept at a field of 1.3 T for 10 s, allowing the ligand to relax at low field in presence of the protein. Afterwards, the solution is injected into the NMR tube (corresponding to an increase of magnetic field to 9.4 T), and the remaining $^{13}C$ polarization is converted to transverse $^{1}H$ magnetization using the INEPT sequence[20]. Then, half of the transverse $^{1}H$ magnetization is stored along $z$ using a $\pi/4$ flip-back pulse, and only the remaining transverse $^{1}H$ magnetization is detected. In order to record the second spectrum, the previously stored $^{1}H$ magnetization is used for a $T_{1\rho}$ experiment.

## Results and discussions
### Precision and repeatability of the $T_1$ measurement in the bullet-DNP setup

The observation of differential relaxation at low field depends on a repeatable hyperpolarization process. In principle, the achieved nuclear spin polarization prior to sample ejection can differ in subsequent experiments. While the sample ejection occurs under computer control with nominally identical settings[21], polarization losses during the transfer and dissolution stages may be sensitive to parameters that can not be controlled with arbitrary precision, such as the helium level inside the polarizer or the fragmentation and mixing of the hyperpolarized sample upon impact of the solid sample on the solution in the solvent reservoir.

In order to test repeatability, we performed five consecutive experiments with nominally identical settings. In each experiment, we hyperpolarized pyruvate (see "Methods"), dissolved it in a protein-free solution, and recorded two proton spectra of the pyruvate methyl signal as described in Fig. 1. The resulting spectra, normalized for the solid-state signal intensity prior to sample ejection, are shown in Fig. 2. In a second normalization step, the five first spectra were normalized to have a mean value of $\langle I_1 \rangle = 1$. The obtained intensities are shown in Fig. 2B and have a standard deviation of

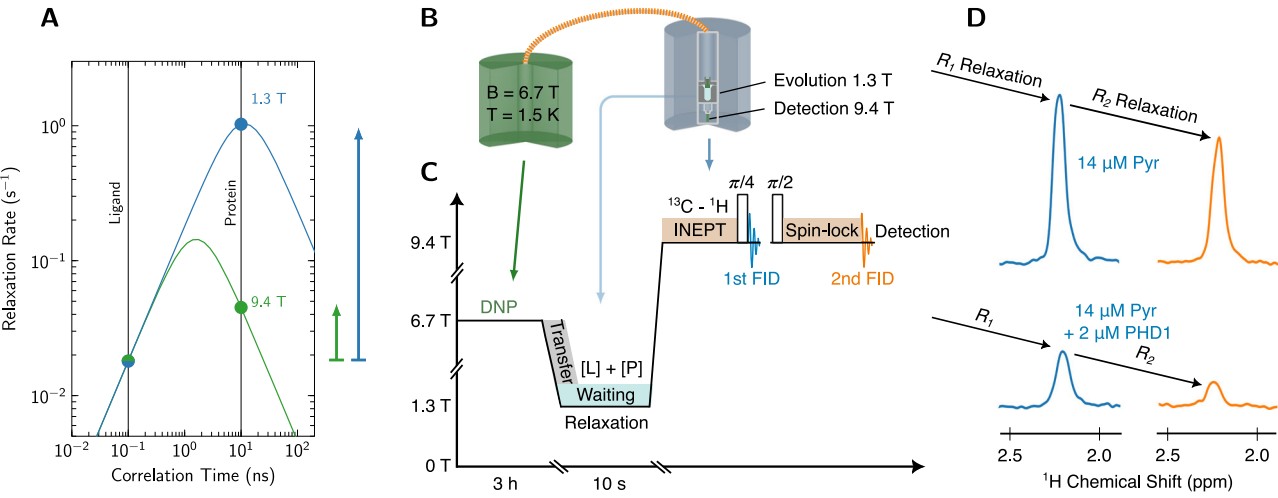

**Fig. 1 | Low-field relaxation for ligand-binding. A** $^{13}C$ relaxation rate for random field relaxation at 9.4 T (green curve) and 1.3 T (blue curve), according to Eq. (1) assuming, for simplicity, that the correlation time of the bound ligand is equal to the correlation time of the protein. The random field amplitude was set to $B_r = 0.2$ mT. The change in relaxation rate is indicated for 9.4 and 1.3 T with green and blue upward arrows, respectively. While the difference in relaxation rate is small at high-field, an approximately 50-fold increase is observed at 1.3 T. **B** Schematic of the experiment with polarization at 6.7 T (green magnet), transfer via a magnetic tunnel (orange), evolution for 10 s at 1.3 T (inside the blue magnet) and subsequent detection at 9.4 T. For a detailed description of the system we refer

to refs. 20,21. **C** Time evolution of the magnetic fields and signal acquisition. DNP (green), transfer process (gray), low-field relaxation (sky blue), signal acquisition via INEPT (blue) and spin-lock (orange). **D** Spectra of the first and second FID for 14 μM pyruvate (top) and 14 μM pyruvate in presence of 2 μM PHD1 protein (bottom). The contrast due to the protein is generated by the waiting time of 10 s at low field, and calculated based on the signal intensity that is available in the first spectrum (blue). Ligand-protein interactions accelerate $R_1$ relaxation, leading to a reduction in the intensity of the first spectrum. The subsequent acquisition of the $R_2$ or $T_{1\rho}$ decay during the spin lock (orange) serves as an independent measurement of binding.

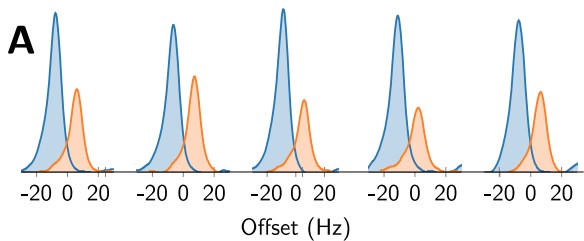

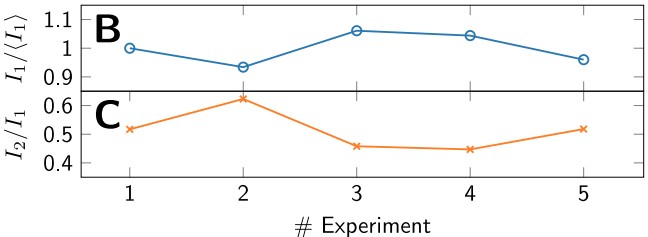

**Fig. 2 | Repeatability of the experiment.** Five nominally identical experiments were carried out. In each experiment, two spectra were recorded as described in Fig. 1. In each experiment, the resulting pyruvate concentration was 14 μM, and the evolution time at 1.3 T was 10 s, followed by detection at 9.4 T. The obtained spectra were normalized by the solid-state signal recorded prior to sample ejection from the polarizer. The methyl part of each spectrum is shown in (**A**). The first (blue) and second (orange) spectra of each experiment are offset for clarity. In **B**, we show the intensities of the first five spectra, normalized by their mean $\langle I_1 \rangle$. The standard deviation of the intensities $I_1/\langle I_1 \rangle$ is 5%. In **C**, we show the intensities of the second five spectra, normalized, respectively, by the intensity of the first spectrum in each experiment. The absolute standard deviation of the intensities $I_2/I_1$ shown in **C** is 6%, the relative standard deviation is 12%.

only 5%. The intensities of the second spectra were normalized individually by the intensities of the first spectra, and are shown in Fig. 2C. In absolute terms, the standard deviation of the second spectra is 6%, i.e. similar to that of the first. Since the mean of the second spectra is only $\langle I_2/I_1 \rangle = 0.51$, the relative standard deviation of the intensities $I_2$ however is 12%.

The high repeatability of the intensities of the first scan indicates that additional relaxation due to the presence of a protein may be observed reliably. Although the low standard deviation of 5% indicates that a binding threshold as high as 90% may be chosen, the repeatability over larger time spans and across changing protein concentrations may be more limited, and we chose to set the threshold to $0.8\langle I_1 \rangle$ for the $T_1$ contrast, and to $0.7\langle I_2/I_1 \rangle$ for the $T_2$ contrast.

## Ligand-binding

As the protein target we choose the prolyl hydroxylase domain 1 (PHD1), a member of the prolyl hydroxylase domain (PHD) protein family. These proteins are involved in the oxygen-dependent degradation of hypoxia-inducible factors (HIFs) and regulate cancer metabolism[22–26]. We note that PHD1 contains a paramagnetic $Fe^{2+}$ center. During purification, the $Fe^{2+}$ was exchanged with $Mn^{2+}$, as in preparatory experiments it was found that relaxation effects were enhanced approximately two-fold with this exchange. This exchange of the metal thus allowed reducing the protein concentration, while, in general, the scaling of relaxation with correlation time should be preserved[27,28].

As a reporter molecule[29] we choose 2-$^{13}$C-pyruvate, for which we have recently described a sensitive $^1$H-based detection scheme that is transferable, e.g., to $^{13}$C-labeled amino acids[20].

The $^1$H signals of the methyl protons of pyruvate, recorded using the pulse sequence shown in Fig. 1, are shown for increasing concentrations of PHD1 protein in Fig. 3. The nominal pyruvate concentration in the final solutions is 14 μM in all experiments. For each protein concentration we calculate a ratio $Q_1$ due to $T_1$ relaxation at low field, and $Q_2$, due to $T_{1\rho}$ relaxation at high field. For each experiment, the first recorded spectrum is shown in blue, and the second spectrum is shown in orange. The second

spectra are offset horizontally for clarity. An exponential line-broadening of 5 Hz has been applied to all spectra, and the signal intensities are calculated as integrals over the frequency range shown in Fig. 3A.

We define the $Q_1$ binding score due to low-field $T_1$ as:

$$Q_1 = \frac{I_1^c}{\langle I_1^{\text{free}} \rangle}, \qquad (2)$$

where $I_1^{\text{free}}$ is the signal intensity of the first spectrum in the absence of protein, and $I_1^c$ is the signal intensity of the first spectrum in the presence of protein, and, if present, competitor.

Similarly, the $Q_2$ score is defined as the ratio of signal intensities of the first and second scan in presence of the protein normalized by the same ratio in absence of the protein, i.e.:

$$Q_2 = \frac{I_2^c}{I_1^c} \bigg/ \frac{I_2^{\text{free}}}{I_1^{\text{free}}}. \qquad (3)$$

Note that $Q_1$ and $Q_2$ take values between 0 and 1 with $Q = 1$ indicating no binding, and $Q < 1$ indicating binding. Competition binding can be observed if the presence of a competitor pushes $Q$ back towards 1.

The spectral intensities of the first two scans are shown for three bullet experiments using protein concentrations of 0 (free), 1 and 2 μM, respectively in Fig. 3A. While no substantial change is observable upon addition of 1 μM PHD1 protein, the $Q_1$ ratio (shown in panel C) is $< 0.5$ upon addition of a total of 2 μM protein. Thus binding can be detected at a low ligand concentration of 14 μM using a $^{13}$C-labeled compound.

Ligands without a $^{13}$C label can now be examined for binding in competition experiments using 2-$^{13}$C-pyruvate as a reporter. Adding 10 μM of the strongly binding competitor 2,4-pyridinedicarboxylic acid (PDCA) prevents the protein from relaxing pyruvate, such that its signal is restored. This is manifested as an increase in $Q_1$ from 0.4 to 0.9 indicated by the upward solid arrow in panel (E). By comparison, the ratio $Q_2$ upon addition of protein (panel D) remains above 0.8 and therefore above the threshold for being significant. Upon adding the competitor, a small reduction in intensity of the second signal is obtained, indicated by a solid downward arrow in panel (F).

This first series of experiments therefore establishes a higher sensitivity to binding events of the $T_1$-based approach, where weak binding and competition effects are manifested in clear and significant changes in signal intensity. In contrast, $T_2$-effects measured in the exact same sample are not above the level of significance.

We decided to repeat the experiments, and data from the second set are shown in panel (B). The experiments are nominally equivalent to those in panel (A), except for the competition experiment, where we chose to add a higher concentration (60 μM) of PDCA in an attempt to observe the competitive binding also on $Q_2$. We note that the relative intensities match the ones in panel (A) well (the small lineshape distortion is due to a technical fault of the shimming system). The competitive binding is again clearly apparent in $Q_1$ by the (dashed) upward arrow in panel (E). With the higher PDCA concentration, also $T_{1\rho}$ is prolonged upon addition of the competitor, indicated by an upward dashed arrow in panel (F).

As discussed in the section on repeatability, we choose to set the thresholds for observing binding at 0.8 for $Q_1$ and at 0.7 for $Q_2$. Using this conservative criterion, we find significant binding in both $T_1$-based experiments that employ a 2 μM protein concentration as well as in both experiments that show competitive binding of PDCA. For the $T_2$-based experiments the data are less clear, since the observed effects are above or close to the threshold values.

In the experiments reported here, the $T_1$-based contrast outperforms the $T_2$-based scheme, while $T_2$-based schemes are used almost exclusively for drug screening in high-field NMR. This does not come as a complete surprise, since the $T_1$-based contrast is only effective at low-field, and field-cycling instrumentation[30] is hence required to observe this

**Fig. 3 | Ligand binding experiments. A** First (blue) and second (orange) spectra at increasing protein concentration, recorded as described in Fig. 1. The protein concentration is indicated for all experiments at the bottom of (**D**), except for the competitive experiments with PDCA, for which it was kept at 2 μM. The pyruvate concentration is 14 μM in all experiments. While only a weak effect is observed when adding 1 μM protein, a substantial effect is observed for 2 μM protein. The signal is nearly restored to full intensity when 10 μM PDCA are added, indicating that PDCA is a strong competitive binder. **B** Shows a repetition of the experiments shown in (**A**), where the PDCA concentration was increased to 60 μM. Solid lines are used for the data shown in (**A**), and dashed lines are used for the data in (**B**). In C, D, we show the ratios $Q_1$ and $Q_2$ for the datasets shown in A and B using solid and dashed lines respectively. Note that we use the same $y$ axis range for both ratios. Error bars are shown with relative errors of 14% for $Q_1$ (accounting for the uncertainty in the measured value as well as in the mean) and 24% for $Q_2$ (2 times the standard deviation of the data shown in Fig. 2). In **E**, **F**, the effect of PDCA on $Q_1$ and $Q_2$ is indicated using an arrow starting from the $Q$ value without competitor, and ending at the $Q$ value that is observed with the competitor. Competitive binding suppresses protein-mediated pyruvate relaxation, and hence leads to an increase of both $Q_1$ and $Q_2$.

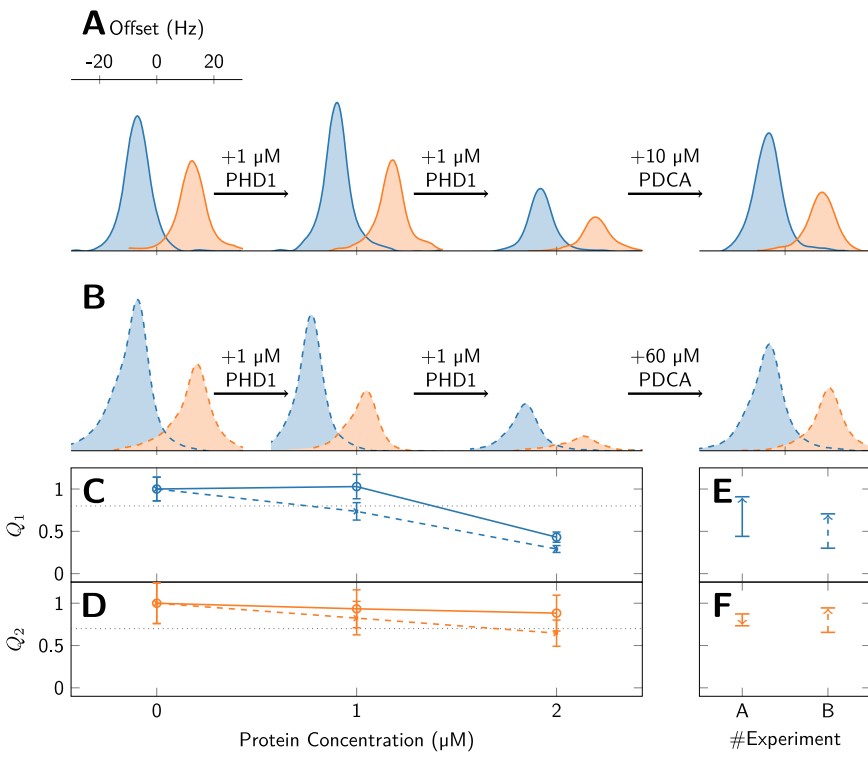

contrast. The $T_1$-based contrast can evolve over very long times (10 s) in our experiments, whereas $T_2$-based schemes are limited to typically 0.3 s by technical limits with respect to the duration and amplitude of radio-frequency pulses. In both sets of experiments the addition of only 1 μM of PHD1 protein does not generate significant contrast in either $Q_1$ or $Q_2$. We provisionally attribute the lack of contrast to adsorption of protein on the ~1 m long capillary that is used to inject the buffer solution into the solvent reservoir. The adsorption of protein likely contributes also to the small but significant differences observed for the $Q$ values in the two sets of experiments.

We note that the $^{13}$C $T_1$-based mechanism reported here should also be observable using protons hyperpolarized by DNP or an alternative means.

Note that we used a total solvent volume of 700 μL, while conventional drug screening experiments use a volume of typically only 200 μL. The reduction in protein concentration described here does therefore not directly translate to a corresponding reduction in protein mass. We plan to extend our work to microfluidic detection schemes with substantially reduced solvent volumes in the near future.

## Conclusion
We have demonstrated that low-field relaxation can be used as a sensitive contrast mechanism for detecting ligand-binding. With hyperpolarized ligands, the ligand signal can be detected with high repeatability at concentrations as low as 14 μM, in turn enabling a reduction in protein concentration to 2 μM. The key limitation of the experiments described here remains the shot-to-shot repeatability of the signal intensity. This requirement for repeatability can be relaxed substantially, if a proper low-field $T_1$ measurement is carried out. Then, $T_1$ itself can be used to calculate contrast[31]. Such a measurement can be done by performing the hyperpolarized NMR measurement in a low-field bench-top system[32], or by storing the hyperpolarized solution at low-field similar to the experiments described here, but transferring sequentially only small aliquots of the hyperpolarized solution to the high-field detection region, where for increased sensitivity the latter experiment can either be carried out with cryoprobes or with

microfluidic detectors[33–35]. We plan to explore both of these approaches in the near future.

## Methods
### Protein preparation
For the expression of PHD1, plasmid DNA (pET-28b(+)) containing a His6-GB1-Presc-PHD1(168--407) construct was transformed into *E. coli* BL21(DE3) Codon Plus RIL cells (Agilent) by heat shock. Isolated colonies were obtained overnight at 37 °C after plating on agar plates supplemented with chloramphenicol (34 μg/mL) and kanamycin (50 μg/mL). Colonies were picked to inoculate an LB preculture which was grown overnight at 30 °C in a shaker. Protein expression was performed in YT 2x medium supplemented with 10 g/L glycerol, 25 mM $KH_2PO_4$, 25 mM $Na_2HPO_4$, 2.5 mM $Na_2SO_4$ and 1000x metal mix (8.33 g/L $FeCl_3 \cdot 6 H_2O$, 0.13 g/L $CuCl_2 \cdot 2 H_2O$, 0.1 g/L $CoCl_2 \cdot 6 H_2O$, 0.84 g/L $ZnCl_2$, 0.01 g/L $H_3BO_3$, 50 g/L ethylenediaminetetraacetic acid (EDTA) pH 8) as well as 10,000x metal mix (33.7 g/L $CuSO_4 \cdot 5 H_2O$, 5 g/L $CoCl_2 \cdot 6 H_2O$, 18 g/L $MnSO_4 \cdot H_2O$, 4.3 g/L $ZnSO_4 \cdot 7 H_2O$) (both inhouse production). All cultures contained 34 μg/mL chloramphenicol and 50 μg/mL kanamycin. Aiming for a starting $OD_{600}$ of 0.1, the respective volume of preculture (typically around 15 mL per 1 L culture) was added to 1 L main cultures in 5 L baffled Erlenmeyer flasks. The cultures were incubated at 30 °C in a shaker until an $OD_{600}$ of 2 was reached. Subsequently, they were cooled to 18 °C for 30 min. Protein expression was induced by addition of 1 mM isopropyl $β$-D-1-thioga-lactopyranoside (IPTG). After expression overnight at 18 °C, cells were harvested the next day by centrifugation for 10 min at 6000 × $g$ and 4 °C.

Cells lysis was performed in lysis buffer (50 mM Tris pH 8.0, 300 mM NaCl, 5 mM imidazole, 1 mM DTT, 1 mM PMSF and 10% (v/v) glycerol) supplemented with DNase I (AppliCHem) with an LM10 Microfluidizer (Microfluidics). The lysate was centrifuged for 45 min at 35,000 × $g$ and 4 °C and the supernatant was vacuum-filtered through a 0.45 μm nitrocellulose membrane (Merck, MF-Millipore). Ni-affinity purification was performed on a 5 mL Ni-NTA Superflow-Cartridge (Qiagen) preequilibrated with washing buffer (50 mM Tris pH 8.0, 300 mM NaCl, 5 mM imidazole, 1 mM DTT, 10% (v/v) glycerol). The protein was step eluted with elution buffer

(50 mM Tris pH 8.0, 300 mM NaCl, 250 mM imidazole, 1 mM DTT and 10% (v/v) glycerol) and the buffer was exchanged back to washing buffer on a HiPrep 26/10 Desalting Column (GE). 3C protease (in-house production) was added and the sample was incubated for 3 h at 4 °C. Protease and cleaved tag were removed by reverse Ni-affinity chromatography on a 5 mL Ni-NTA Superflow-Cartridge (Qiagen) preequilibrated with washing buffer. The protein sample was supplemented with 2 mM $MnCl_2$ and incubated overnight at 4 °C to replace $Fe^{2+}$ in the active site by $Mn^{2+}$. Afterwards, the sample was concentrated to approximately 250 μM in a 10 kDa Amicon-Ultra 15 mL centrifugal filter (Merck Millipore) and purified by size exclusion chromatography on a Superdex 75 Increase 16/600 column (GE) to finally obtain the protein in 50 mM Tris pH 7.5, 300 mM NaCl and 1 mM DTT. After concentrating in a 10 kDa Amicon-Ultra 15 mL centrifugal filter (Merck Millipore) to around 100 μM, the protein was aliquoted, flash-frozen and stored at −80 °C.

## Bullet-DNP measurements

All experiments employed a final pyruvate concentration of 14 μM, and were conducted as follows:

For DNP, we used a stock of 5 mM of 2-$^{13}$C labeled pyruvate in $D_2O$ to DMSO (2:1 v:v), with doubly labeled DMSO-$^{13}C_2$ added at a concentration of 1 M, and 15 mM of trityl radical OX063. The bullets were prepared using three layers. First, a 5 μL bead of $D_2O$:Glycerol (1:1) was frozen in liquid nitrogen and inserted into the bullet. Second, a 2 μL frozen bead of the pyruvate solution was added; and third, another 10 μL of the same mixture as in the first layer were added to protect from heat.

The bullets were then loaded into a home-built bullet-DNP polarizer (described in detail in ref. 21 with a pinch valve modification as described in ref. 20). The frozen solid inside the bullet was hyperpolarized at a temperature of typically 1.5 K for typically 2 h, and the buildup was monitored by solid-state NMR using small flip-angle pulses.

Prior to each shot, the solvent reservoir of the injection device inside the liquid-state NMR magnet was filled with a 700 μL solution of buffer (50 mM Tris pH 7.5 300 mM NaCl), resulting in a final concentration of 14 μM in each shot. For experiments with protein, 1 μM or 2 μM of PHD1 were added to the 700 μL solution that was used for dissolving the hyperpolarized sample. In all experiments, the hyperpolarized bullet was shot into the injection device as usual, but the hyperpolarized solution was then kept inside the solvent reservoir for 10 s to allow for relaxation at 1.3 T. The solution is kept at ambient temperature in a titanium reservoir which provides a high heat capacity and a high thermal conductivity. Subsequently the sample was injected into the NMR tube, and the pulse sequence shown in Fig. 1 was used to record the two NMR spectra with information on $R_1$ and $R_2$. For the $T_{1\rho}$ measurement, the strength and duration of the spin-lock field were set to 16.7 kHz and 300 ms, respectively. For competitive binding experiments the specified concentration of PDCA was added to the 700 μL Tris buffer that was used for dissolving the hyperpolarized sample.

## Reporting summary

Further information on research design is available in the Nature Portfolio Reporting Summary linked to this article.

## Data availability

The NMR raw data and processing files are available from `KITopen` at https://doi.org/10.35097/4df4g6du1awng50h (ref. 36).

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

## Acknowledgements
This work has received funding from the European Research Council (ERC) via the Synergy grant "Highly Informative Drug Screening by Overcoming NMR Restrictions" (HiSCORE, grant agreement no. 951459), from the "Impuls- und Vernetzungsfonds of the Helmholtz-Association" (grant number VH-NG-1432), and from the Deutsche Forschungsgemeinschaft (DFG, grant number 454252029 - SFB1527).

## Author contributions
P.N. conducted the experiment with support from M.M. and N.L. P.N. analyzed data with support from B.M. A.G., B.M., and P.N. conceived the experiment. N.L. produced the PHD protein, and N.L., A.G., and P.N. wrote and adapted the pulse sequence. B.M. and P.N. wrote the manuscript with contributions by A.G. and N.L. All authors reviewed the manuscript.

## Funding

## Competing interests
All authors have filed an EU patent application on drug screening using DNP and low-field relaxation as a contrast mechanism (EP 25152466.6). B.M. is co-owner of HyperSpin Scientific UG.
