## [Transparent Peer Review file · Communications Chemistry]

Single-Scan Detection of Ligand-Binding using Hyperpolarization and Low-field Relaxation

Corresponding Author: Dr Benno Meier

Version 0:

Reviewer comments:

Reviewer #1

(Remarks to the Author)

The authors present an NMR method to utilize longitudinal relaxation (T1) differences to probe for small molecule – protein target interactions. To ensure sufficient signal intensity and to enhance the contrast based on T1 relaxation for observing binding of a small molecule ligand to a target protein they combine hyperpolarization and low-field relaxation. To enable this a “bullet-DNP” polarizer and q field-cycling NMR instrumentation setup has been developed.

The authors first review the current methods of ligand-observed NMR and introduce the theoretical concept of their work. Then the reproducibility of ligand signal intensities are assessed when using their bullet-DNP technique, defining the thresholds for interaction studies. The method is tested with the binding of ^{13}C -pyruvate to PHD1 at two different concentrations, as well as displacement of pyruvate by PDCA. The main finding of the study is that T1 can be used to observe ligand binding, and can outperform T2 based measurements.

The manuscript presents an interesting novel conceptual idea for ligand screening. However, the requirement of a bullet DNP setup with long durations for low field relaxation, and observation of a ^{13}C -labeled ligand renders this technically highly complex and demanding. A general and broad applicability is not given. This is a proof-of-concept study that explores an interesting concept to achieve T1 based contrast in observing protein binding by small molecule ligands.

The manuscript can be published but the authors should address and clarify the points below.

Main comments:

- The title’s catch-phrase “Single-Scan Detection” does not seem to be the focus of the manuscript. It is clear from the lengthy setup that only one scan is possible, so it appears to be more a bug than a feature. The novelty and focus of the manuscript is the demonstration that T1 relaxation can be used to monitor small molecule/protein binding. The title should be adapted to reflect the key result.
- It should be clarified from the beginning that the presented approach to study small molecule binding is based on displacement of a reporter molecule.
- The results are based on duplicates of one binding study (2- ^{13}C pyruvate reporter molecule) and a displacement (with PDCA). Additional displacement experiments would underline the robustness of the technique.
- The Q values of the repetition experiment differ significantly from the ones of the first one: In the second experiment Q1 is already below the threshold at $1\mu\text{M}$ PHD1 and Q2 is below the threshold at $2\mu\text{M}$ PHD1, contrary to the results of the first experiment. This is confusing since the thresholds were conservatively estimated from independent experiments in the first Results section and Figure 2. Are the thresholds flawed? Is this due to the shimming problem mentioned? Unfortunately, the authors do not comment on this, this should be explained.

Minor comments:

- The instrumental setup is not entirely clear to me. Figure 1B and its figure text, as well as the Methods section, could be improved to give a better understanding of the setup, without the need to dive into the cited literature.
- The acronym PDCA should be written with its full name at least once
- What is meant with “The second spectra are offset horizontally for clarity.” (line 92). This sentence should be stated in the Figure 3 legend and not into the main text to avoid confusion.
- The abstract conclusions highlight that only an amount of $2\mu\text{M}$ protein per sample is needed, however the sample size of $700\mu\text{M}$ is at least four-times larger than in a conventional ligand-observed NMR experiment.
- It is unclear why Fe^{2+} was replaced with also paramagnetic Mn^{2+} , and not for instance Mg^{2+} .

Reviewer #2

(Remarks to the Author)

NMR spectroscopy is a highly valuable tool for the screening of binding of ligands of low molecular weight to biomacromolecular targets (proteins, RNA, DNA). Depending on the affinity regime of the studied interaction, different NMR methodologies can be applied. Common limitations for NMR are low sensitivity and, linked to this, the range of interactions that can be faithfully determined. While NMR is strong in detecting weak interactions, it has its limits in detecting strong interactions, as typically at least 10 micromolar samples are required to obtain an NMR signal in a meaningful time frame. The approach pursued here is the development of a top-level new methodology that uses DNP enhanced increase of signal-to-noise by hyperpolarization technology combined with low-field T1 relaxation weighing. Samples are hyperpolarized by hyperpolarized NMR, transferred to a low field magnet where T1 differences between free ligand (low molecular weight molecule) and bound ligand (high molecular weight ligand-target complex) are substantially larger than at high field and then subsequent transfer to high magnetic field to capitalize on its high detection sensitivity and high chemical shift dispersion.

The authors first optimized their approach with a ^{13}C labeled ligand. While this approach is elegant in principle, its general applicability is limited, as chemical synthesis of ^{13}C ligands is time-consuming.

In order to avoid this, the authors also conduct competition experiments. In such competition experiments, only a single reference compound needs to be labeled, and evolution of affinity can be assessed over a series of follow-up compounds. In general, I recommend publication of the manuscript. There is a single suggestion that I wish to make that the authors can consider in the final manuscript.

1.) For competition experiments to work, the relative K_D 's between reference compound and tested compound must not be too different. The authors should comment and/or cite literature on this aspect.

Reviewer #3

(Remarks to the Author)

This manuscript presents a methodological study focused on the analysis of ligand-protein interactions using dissolution Dynamic Nuclear Polarization (dDNP) in the bullet-DNP variant. The authors describe a protocol involving the hyperpolarization of a ligand followed by its mixing with a protein. While the general approach of using dDNP for such studies is not novel, the authors innovatively exploit the contrast provided by low magnetic fields to leverage T1 relaxation as a means of increasing the contrast for the ligand-protein interactions. This methodological refinement represents a notable contribution to the field, and I believe that this is the right way of doing it.

The manuscript is well-structured and clearly written, with informative and accessible figures. However, I note the absence of raw data, particularly regarding the polarization measurements and the underlying dataset, which would strengthen the reproducibility and transparency of the study.

I read your manuscript with great interest. As you probably know, I am quite familiar with this field and I hope to provide constructive feedback to further improve an already excellent piece of work. I genuinely believe that this topic has outstanding potential for both the dDNP and biological communities, as it directly addresses the limitations in protein production and enhances our ability to map strong binding events.

A recent article from my former group was published on a related topic and, in my opinion, deserves to be cited. Nevertheless, I find that your present version represents a significant improvement for several reasons. The mixing between the protein and the ligand is performed at low magnetic field, which is indeed the appropriate way to maximize contrast – while acquisition occurs at high field, which will ultimately be essential for practical applications. Furthermore, bullet-DNP is probably a more reliable approach than dDNP.

In the conclusion, I would recommend briefly discussing the work on 2F-NMR introduced by Ferrage et al. and eventually relaxometry (high-resolution or field cycled). This would add depth to your discussion of the project's potential: by controlling polarization at around 1.4 T, one could improve relaxation rate measurements and develop a powerful approach for investigating protein-ligand dynamics.

I have suggested a few specific improvements below. Some are minor remarks, while others may help guide future developments in this promising direction

Line 22 – 23:

“In summary, the established ... do not make use of T1 relaxation.”

This assumption completely overlooks the potential of relaxometry to map protein-ligand interactions (see, for instance, the work of A. Yurovskaya)

Line 37-38

“Using hyperpolarization techniques such as dDNP ...”

I suggest citing the work performed by my former group (DOI: [10.1021/acsomega.4c05101]). Even though the version proposed here employs high-field acquisition combined with bullet-DNP – which I consider the most appropriate approach – referencing this study would provide a useful contextual link.

Line 46-47 (equation 1)

This equation is the model for the longitudinal relaxation time, and is highly simplify, even if I believe that it's the most appropriate to use. I would recommend clarifying the underlying assumptions – in particular stressing that relaxation takes place at 1.4 T where relaxation equation are still 100 % valid.

Line 60

“ Prior to detection ... is then subject to a T1ρ experiment.”

This formulation is somewhat confusing. I would suggest rephrasing it for clarity.

Line 69 (paragraph)

I fully agree with this systematic verification. Would it be possible to first record the signal and then mix with the protein to (1) check the sample temperature beforehand and (2) account for the t=0 polarization level?

Temperature is a crucial aspect of protein ligand binding... You don't specify at which temperature the mixing occurs.

Figure 3:

Are the error bars displayed but overlapping with the data points? The signal for the 2 μM case appears particularly weak, and it's associated uncertainty may be larger than for the other concentrations.

Line 151:

“and 1000x as well as 10 0000x metal mixes (inhouse production)”.

Some precisions are needed.

Line 175 :

I'm surprised that you did not put 4/6 ratio in D2O/Glycerol (I guess it's volumic ratio (to specify). There is the article of Songi Han showing the polymorphism at this concentration...

Line 186:

T1rho experiment does not specify the RF field... which is important for the reproducibility of the curve.

Overall, this is a very strong and promising manuscript. I am confident that addressing these minor points will make it even more impactful.

Reviewer : Samuel Cousin

Version 1:

Reviewer comments:

Reviewer #1

(Remarks to the Author)

The authors have responded and mostly clarified the points raised by myself and the other reviewers, so I support publication of the manuscript.

Reviewer #2

(Remarks to the Author)

The authors decided not to change anything.

My statement that displacement does not always work is valid. I assume that the authors have not conducted such studies themselves.

While I was positive in the first round of reviews, the attitude of the authors not to change anything of any of the reviewer's valid comments does not allow to recommend publication of the article in the current form.

Reviewer #3

(Remarks to the Author)

I am satisfied with the corrections provided by the authors. The manuscript has been revised appropriately and, in its current form, is suitable for publication.

Reviewer #1

1. The title's catch-phrase "Single-Scan Detection" does not seem to be the focus of the manuscript. It is clear from the lengthy setup that only one scan is possible, so it appears to be more a bug than a feature. The novelty and focus of the manuscript is the demonstration that T1 relaxation can be used to monitor small molecule/protein binding. The title should be adapted to reflect the key result.

Title : Single-Scan Detection of Ligand-Binding at Low Micromolar Concentrations using Hyperpolarization and Low-field Relaxation

We prefer not to change the title. While the referee has a valid point given the long polarization times, our lab has made substantial progress towards fully automated drug screening at a repetition rate < 1 h, and with substantial potential for further acceleration.

2. It should be clarified from the beginning that the presented approach to study small molecule binding is based on displacement of a reporter molecule.

We note that we made reference to a ^{13}C labeled reporter ligand in the abstract. We are however willing to expand the role of the ^{13}C ligand in the final sentence of the abstract (changes in green):

The scheme is expanded to a label-free ligand via a competitive binding experiment in which the label-free ligand displaces the ^{13}C labeled reporter ligand.

3. The results are based on duplicates of one binding study (2- ^{13}C pyruvate reporter molecule) and a displacement (with PDCA). Additional displacement experiments would underline the robustness of the technique.

The goal of this study is to introduce and demonstrate the feasibility of the methodology rather than to conduct an extensive drug-screening assay. While we are working towards fully automated screening experiments, we believe the presented binding and displacement data are sufficient to validate the concept.

4. The Q values of the repetition experiment differ significantly from the ones of the first one: In the second experiment Q1 is already below the threshold at $1\ \mu\text{M}$ PHD1 and Q2 is below the threshold at $2\ \mu\text{M}$ PHD1, contrary to the results of the first experiment. This is confusing since the thresholds were conservatively estimated from independent experiments in the first Results section and Figure 2. Are the thresholds flawed? Is this due to the shimming problem mentioned? Unfortunately, the authors do not comment on this, this should be explained.

We thank the referee for bringing this point to our attention. As discussed in the submitted version of this article, the addition of only $1\ \mu\text{M}$ of PHD1 protein does not generate significant contrast. We attributed this observation to adsorption of protein on the capillary used to inject the buffer solution. We suggest to extend the passage as follows:

We provisionally attribute the lack of contrast to adsorption of protein on the $1\ \text{m}$ long capillary that is used to inject the buffer solution into the solvent reservoir. The

adsorption of protein likely contributes also to the small but significant differences observed for the Q values in the two sets of experiments.

Minor comments

- The instrumental setup is not entirely clear to me. Figure 1B and its figure text, as well as the Methods section, could be improved to give a better understanding of the setup, without the need to dive into the cited literature.

The setup is indeed not trivial. We have added arrows linking the events in panel C to the magnets in which they occur (panel B). We have modified the caption of Figure 1B and added two references that specify the hardware exactly:

Low-field relaxation for ligand-binding. (A) ^{13}C relaxation rate for random field relaxation at 9.4 T (green curve) and 1.3 T (blue curve), according to Eq. 1 assuming, for simplicity, that the correlation time of the bound ligand is equal to the correlation time of the protein. The random field amplitude was set to $B_r = 0.2\text{mT}$. We note that a similar dependence is obtained for dipole-dipole relaxation. While the difference in relaxation rate is small at high-field, an approximately 50-fold increase is observed at 1.3 T. (B) Schematic of the experiment with polarization at 6.7 T (green magnet), transfer via a magnetic tunnel (orange), evolution for 10 s at 1.3 T (inside the blue magnet) and subsequent detection at 9.4 T. For a detailed description of the system we refer to Refs. XX, YY. (C) Time evolution of magnetic fields and signal acquisition: DNP (green), transfer process (grey), low-field relaxation (sky blue), signal acquisition via INEPT and spin-lock. (D) Spectra of the first and second FID for free pyruvate (top) and pyruvate in presence of 2 μM PHD1 protein (bottom). The contrast due to the protein is generated by the waiting time of 10 s at low field, and calculated based on the signal intensity that is available in the first spectrum (blue). Ligand-protein interactions accelerate R_1 , giving rise to a positive ΔR_1 . The subsequent acquisition of the R_2 or $T_{1\rho}$ decay during the spin lock (orange) serves as an independent measurement of binding.

- The acronym PDCA should be written with its full name at least once

The relevant sentence now reads:

Adding 10 μM of the strongly binding competitor 2,4-pyridinedicarboxylic acid (PDCA) prevents the protein from relaxing pyruvate, such that its signal is restored.

- What is meant with “The second spectra are offset horizontally for clarity.” (line 92). This sentence should be stated in the Figure 3 legend and not into the main text to avoid confusion.

We agree. We have moved the sentence to the caption of Fig. 3.

- The abstract conclusions highlight that only an amount of 2 μM protein per sample is needed, however the sample size of 700 μM is at least four-times larger than in a conventional ligand-observed NMR experiment.

We thank the referee for raising this issue. While we agree, we see the sample volume rather as an opportunity for further reduction. We have added the following comment to the discussion:

Note that we used a total solvent volume of 700 μL , while conventional drug screening experiments use a volume of typically only 200 μL . The reduction in protein concentration described here does therefore not directly translate to a corresponding reduction in protein mass. We plan to extend our work to microfluidic detection schemes with substantially reduced solvent volumes in the near future.

- It is unclear why Fe^{2+} was replaced with also paramagnetic Mn^{2+} , and not for instance Mg^{2+} .

We thank the referee for also paying attention to the details of protein preparation. The protein is typically the most expensive reagent in the experiment. Therefore, we aimed at increasing the relaxation effect caused by the protein by exchanging the Fe^{2+} with the paramagnetic Mn^{2+} . The plan was to reduce the protein concentration. Experimentally, exchanging Fe^{2+} with Mn^{2+} approximately doubles the relaxation effect, such that the protein concentration can be reduced by factor 2. This was determined in preliminary experiments. Interestingly, Fe^{2+} can be low spin or high spin ($S = 2$) depending on the ligand field. In PHD1 the iron is complexed by ligands inducing a weak ligand field effect, i.e. with His, Asp/Glu and alpha ketoglutarate, (or pyruvate in our experiment), such that a high spin state would be expected. Mn^{2+} is expected to always be in the high spin state ($S = 5/2$), such that replacement of Fe^{2+} with Mn^{2+} is expected to result in an increased relaxation effect, as we observed in our control experiments. Replacement of Fe^{2+} with Mg^{2+} or Zn^{2+} to obtain a stably diamagnetic species, would increase the required protein concentration and was therefore not pursued. The sentence where the protein is described was now changed in the following way:

We note that PHD1 contains a paramagnetic Fe^{2+} center. During purification, the Fe^{2+} was exchanged with Mn^{2+} , as in preparatory experiments it was found that relaxation effects were enhanced approximately two-fold with this exchange. This exchange of the metal thus allowed reducing the protein concentration, while, in general, the scaling of relaxation with correlation time should be preserved.^{24,25}

Reviewer 2

1. For competition experiments to work, the relative K_D 's between reference compound and tested compound must not be too different. The authors should comment and/or cite literature on this aspect.

The referee's point is valid when one seeks to determine K_D . In a competitive binding study, a high-affinity ligand can always be found, even if the affinity of the reporter ligand is much smaller. We have added a citation to a detailed review on affinity measurements, written by one of us (Alvar Gossert) in line 86:

As a reporter molecule^{XX} we choose...

Reviewer 3

1. Line 22 – 23:

“ In summary, the established ... do not make use of T1 relaxation. “This assumption completely overlooks the potential of relaxometry to map protein-ligand interactions (see, for instance, the work of A. Yurovskaya)

We believe that the referee refers to the recent work by Yurkovskaya. While this work is intriguing, it does require the existence of an accessible long-lived spin state on the ligand. We have added the following statement to the text (after line 43):

We note that long-lived spin states are often highly sensitive to binding and can be used to detect binding at extremely low protein concentrations. Examples of ligands that exhibit such states are molecules with aliphatic chains^{XX} and alanine-glycine and citrate^{XX}.

2. Line 37-38

“ Using hyperpolarization techniques such as dDNP ...”I suggest citing the work performed by my former group(DOI:[10.1021/acsomega.4c05101]). Even though the version proposed here employs high-field acquisition combined with bullet-DNP – which I consider the most appropriate approach – referencing this study would provide a useful contextual link.

We thank the referee for bringing this work to our attention. We now cite and discuss this work in the introduction (after line 40):

The group of Jannin has recently combined hyperpolarization with benchtop NMR and was able to observe binding of hyperpolarized ligands at a concentration of 600 μM using a protein concentration of 40 μM .^{XX}

3. Line 46-47 (equation 1)

This equation is the model for the longitudinal relaxation time, and is highly simplify, even if I believe that it's the most appropriate to use. I would recommend clarifying the underlying assumptions – in particular stressing that relaxation takes place at 1.4 T where relaxation equation are still 100 % valid.

We agree. We have moved one sentence from the caption of Fig. 1, and append this to the paragraph ending on line 52.

We note that a similar dependence of relaxation rate on correlation time is obtained for dipole-dipole relaxation. Other relaxation mechanisms (such as chemical shift anisotropy) are assumed to be negligible.

4. Line 60

“ Prior to detection ... is then subject to a T1 ρ experiment.” This formulation is somewhat confusing. I would suggest rephrasing it for clarity.

We have rephrased as follows:

is converted to **transverse** ^1H magnetization using the INEPT sequence.¹⁸ **Then**, half of the **transverse** ^1H magnetization is stored along z using a $\pi/4$ flip-back pulse, and only the remaining transverse ^1H magnetization is detected. In order to **record the second spectrum**, the previously stored ^1H z-magnetization is **used for** a T1 ρ experiment

5. Line 69 (paragraph)

I fully agree with this systematic verification. Would it be possible to first record the signal and then mix with the protein to (1) check the sample temperature beforehand and (2) account for the $t=0$ polarization level? Temperature is a crucial aspect of protein ligand binding... You don't specify at which temperature the mixing occurs.

This would be possible in theory, but such a procedure would also add complexity and reduced sensitivity due to the polarization loss during sample movement and mixing after the $t=0$ polarization level measurement.

We now state the temperature of mixing in the section on Bullet-DNP measurements, line 184:

... to allow for relaxation at 1.3 T. **The solution is kept at ambient temperature in a titanium reservoir which provides a high heat capacity and a high thermal conductivity.**

6. Figure 3: Are the error bars displayed but overlapping with the data points? The signal for the 2 μM case appears particularly weak, and it's associated uncertainty may be larger than for the other concentrations.

The 2 μM signal is weak due to the significant relaxation as a consequence of binding. For clarity we have now added errorbars to the graph, and the following sentence to the caption of Fig. 3:

Error bars are shown with relative errors of 14% for Q1 (accounting for the uncertainty in the measured value as well as in the mean) and 24% for Q2 (2 times the standard deviation of the data shown in Fig. 2).

7. Line 151:

“and 1000x as well as 10 0000x metal mixes (inhouse production)”. Some precisions are needed.

We added a detailed description of the compositions, and the relevant passage now reads:

and 1000x metal mix (8.33 g/L $\text{FeCl}_3 \cdot 6 \text{H}_2\text{O}$, 0.13 g/L $\text{CuCl}_2 \cdot 2 \text{H}_2\text{O}$, 0.1 g/L $\text{CoCl}_2 \cdot 6 \text{H}_2\text{O}$, 0.84 g/L ZnCl_2 , 0.01 g/L H_3BO_3 , 50 g/L ethylenediaminetetraacetic acid (EDTA) pH 8) as well as 10 0000x metal mix (33.7 g/L $\text{CuSO}_4 \cdot 5 \text{H}_2\text{O}$, 5 g/L $\text{CoCl}_2 \cdot 6 \text{H}_2\text{O}$, 18 g/L $\text{MnSO}_4 \cdot \text{H}_2\text{O}$, 4.3 g/L $\text{ZnSO}_4 \cdot 7 \text{H}_2\text{O}$) (both inhouse production)

8. Line 175 :

I'm surprised that you did not put 4/6 ratio in D2O/Glycerol (I guess it's volumic ratio

(to specify). There is the article of Songi Han showing the polymorphism at this concentration...

In our previously reported work, we optimized the ^{13}C polarization using ^{13}C labeled DMSO, and we chose to use the same composition ($\text{D}_2\text{O}:\text{DMSO}$, 2:1, 2 μL).

9. Line 186: T1rho experiment does not specify the RF field... which is important for the reproducibility of the curve.

We now add this information:

The strength of the spin-lock field was set to 16.7 kHz.